# First-Year Implementation of the EXercise for Cancer to Enhance Living Well (EXCEL) Study: Building Networks to Support Rural and Remote Community Access to Exercise Oncology Resources

**DOI:** 10.3390/ijerph20031930

**Published:** 2023-01-20

**Authors:** Chad W. Wagoner, Julianna Dreger, Melanie R. Keats, Daniel Santa Mina, Margaret L. McNeely, Colleen Cuthbert, Lauren C. Capozzi, George J. Francis, Linda Trinh, Daniel Sibley, Jodi Langley, Joy Chiekwe, Manuel Ester, Aude-Marie Foucaut, S. Nicole Culos-Reed

**Affiliations:** 1Faculty of Kinesiology, University of Calgary, Calgary, AB T2N 1N4, Canada; 2School of Health and Human Performance, Faculty of Health, Dalhousie University, Halifax, NS B3H 4R2, Canada; 3Department of Medicine, Division of Medical Oncology, Nova Scotia Health, Halifax, NS B3H 2Y9, Canada; 4Faculty of Kinesiology and Physical Education, University of Toronto, Toronto, ON M5S 2W6, Canada; 5Department of Physical Therapy, University of Alberta, Edmonton, AB T6G 2G4, Canada; 6Supportive Care Services, Cancer Care Alberta, Edmonton, AB T5J 3E4, Canada; 7Faculty of Nursing, University of Calgary, Calgary, AB T2N 4V8, Canada; 8Department of Clinical Neurosciences, University of Calgary, Calgary, AB T2N 4N1, Canada; 9Health Educations and Promotion Laboratory, UR 3412, University Sorbonne Paris North, F-93000 Bobigny, France

**Keywords:** physical activity, exercise, cancer, online exercise, network development, RE-AIM, underserved, healthcare provider, qualified exercise professional

## Abstract

Barriers to exercise-oncology programs remain for those living with and beyond cancer in rural and remote communities, including geographic isolation and access to programs. The EXercise for Cancer to Enhance Living Well (EXCEL) study was designed to support exercise-oncology implementation in rural and remote communities across Canada. The purpose of this analysis was to evaluate the first-year reach, adoption, and implementation of the EXCEL study. Reach outcomes included participant characteristics, study enrolment, and referral type (self vs. healthcare-provider [HCP] referral). Adoption outcomes included the number of clinical contacts, trained qualified exercise professionals (QEPs), and QEPs delivering EXCEL exercise classes. Implementation outcomes included retention, adherence, assessment completion rates, and adverse-event reporting. A total of 290 individuals living with cancer enrolled in EXCEL in year one, with an 81.4% retention to the study intervention. Most participants self-referred to EXCEL (75.8%). EXCEL’s HCP network consisted of 163 clinical contacts, and the QEP network included 45 trained QEPs, 22 of whom delivered EXCEL classes. Adherence to the exercise intervention was 78.2%, and only one adverse event (mild) was reported. Fitness assessment and patient-reported outcome completion rates were above 85% pre- and post-intervention. EXCEL has developed HCP and QEP networks supporting exercise referral and online delivery, and the intervention is meeting feasibility markers. These implementation findings will inform the continued gathering of feedback across stakeholders to ensure that best evidence informs best practices.

## 1. Introduction

Exercise is an evidence-based intervention that improves psychosocial outcomes, physical fitness and function, and overall quality of life across cancer populations [1]. Despite the growing evidence and promotion of exercise, many individuals living with and beyond cancer remain inactive [2,3]. Specifically, there is a need to reach individuals living with and beyond cancer in rural and remote communities, where barriers such as geographic isolation and lack of awareness of, or access to, exercise-oncology services may contribute to the decreased adoption of a physically active lifestyle [4,5,6]. To address this inequity in access to exercise-oncology resources between individuals living with cancer in urban versus rural and remote communities, accessible exercise resources that can be feasibly and sustainably implemented are needed. Online exercise programs have recently become an alternative to in-person exercise. Preliminary evidence shows online exercise delivery for those living with cancer is safe, feasible, and can improve physical fitness, physical function, body composition, and quality of life [7,8,9,10,11]. Importantly, the shift to online delivery may also provide a unique opportunity to expand outreach to those who do not have access to exercise-oncology resources, including those in rural and remote communities. However, few studies have focused on the accessibility and implementation of online exercise-oncology interventions, particularly in rural and remote populations [12,13]. 

To build the sustainable and effective implementation of exercise oncology resources, the engagement of key stakeholders using an integrated knowledge-translation approach is necessary. Healthcare providers (HCPs) and qualified exercise professionals (QEPs) are key stakeholders responsible for “bridging the gap” from the clinical to community settings. This may be even more crucial when we consider how to best address barriers to better support participants from rural and remote communities [14]. Implementation efforts for exercise-oncology resources targeting rural and remote individuals living with and beyond cancer must include QEP training, building program support and referrals from HCPs, and developing sustainable, community-based fitness partnerships [12]. The EXercise for Cancer to Enhance Living Well (EXCEL) effectiveness-implementation study [15] aims to build these networks with HCP and trained fitness professionals, utilizing a “Hub and Spoke Framework” (see Figure 1), to support sustainable implementation. This approach utilizes clinical exercise physiologists (CEPs) or registered kinesiologists in the “hub” urban centres to build both clinical and fitness networks (i.e., HCP and QEP networks) across rural and remote communities in the “spokes.” In addition, the hub CEP supports the provision of exercise-oncology education and training to HCPs and QEPs.

To support and assess EXCEL implementation, the RE-AIM framework [16,17] is used. RE-AIM provides key implementation factors to assess at both the individual (reach and effectiveness) and organizational levels (adoption, implementation, and maintenance), which may help to identify both facilitators and barriers that impact the effective and sustainable implementation of the evidence-based exercise-oncology intervention. Thus, the purpose of this analysis is to report on the first-year implementation of the EXCEL study as guided by the RE-AIM framework. This evaluation provides insight into EXCEL’s initial implementation efforts for QEP and HCP network development to support the delivery of the online exercise-oncology intervention.

## 2. Materials and Methods

### 2.1. Study Design 

EXCEL is a multisite, five-year, hybrid effectiveness–implementation study and includes assessments at baseline, post-intervention (12-week), and multiple follow-up timepoints, including twenty-four weeks and yearly up to five years post-intervention completion. The present paper focuses on select RE-AIM implementation factors at baseline and twelve-week timepoints during the first year of implementation. Specific details regarding the entirety of the EXCEL protocol have been described previously [15]; it builds on the successful Alberta Cancer Exercise hybrid effectiveness–implementation study [18]. Ethics approval was received from the Health Research Ethics Board of Alberta (HREBA.CC-20-0098), as well as the respective hub ethics boards in Nova Scotia (Halifax) and Ontario (Toronto). The sample presented in the present paper includes data on study outreach, recruitment, and participants who completed EXCEL’s twelve-week exercise-oncology intervention within its first year of implementation (Fall 2020–Fall 2021). 

### 2.2. Participants

Eligibility for EXCEL included those who were 18 years or older living with and beyond cancer, those who were pre-treatment, currently receiving treatment, or post-treatment (up to three years), able to participate in mild levels of physical activity, living in underserved rural/remote communities (population < 100,000 people) [19], and had access to reliable internet for online delivery. Due to the COVID-19 pandemic, EXCEL included individuals living with and beyond cancer from urban areas who lost access to exercise-oncology programs due to pandemic-related restrictions. These urban-based participants are included in the current implementation analyses.

Participants could either self-refer or be referred by a HCP. Screening for exercise participation was conducted online by hub CEPs and included a detailed medical history (cancer-related, treatment-related, and other chronic conditions and injuries) and a physical-activity-readiness assessment via the Physical Activity Readiness Questionnaire+ (PARQ+) [20]. Participants were recruited from three “hubs” (Figure 1) located across Canada: Alberta, Nova Scotia, and Ontario. Recruitment in Fall 2020 included participants from Alberta only; Winter 2021 included participants from Alberta and Nova Scotia; and Spring/Fall 2021 included participants from Alberta, Nova Scotia, and Ontario. Participants provided written informed consent prior to participation.

### 2.3. EXCEL Exercise-Oncology Program

The entirety of the EXCEL program in year one was completed online via ZOOM™. EXCEL employed an “Exercise and Educate” model to provide exercise-oncology behaviour-change support to participants through handouts, a QEP-facilitated discussion of each topic, and optional educational exercise webinars throughout the 12-week exercise intervention. Classes meet twice per week and were 60 min in duration. Each exercise class included a warm-up (5–10 min), circuit-style, multi-modal exercise training consisting of aerobic, resistance, and balance exercises (45–50 min), and a cool-down consisting of full-body flexibility training (5–10 min). Program starts took place in January, April, and September of each year. Class sizes ranged from eight to fifteen participants to ensure individual attention and feedback could be provided as necessary. Fitness assessments and patient-reported outcome measures, both of which assessed the effectiveness of the exercise program, were completed online at baseline and 12-week timepoints. 

### 2.4. RE-AIM Outcomes

The primary outcomes in EXCEL’s year-one implementation analysis focused on the reach, adoption, and implementation of the study. Given the limited work to date examining the online delivery to rural and remote individuals living with and beyond cancer, no a priori levels across implementation markers were set. The effectiveness and maintenance components of RE-AIM were the primary outcomes for the full trial and are not included in the current analysis. As per the study protocol, these primary outcomes will be assessed upon data completion of the study. 

#### 2.4.1. Reach 

The reach of the EXCEL study was assessed through participant characteristics, study enrolment (*n*), reasons for study refusal, referral type, and referral resources, as reported by participants. Characteristics of study participants were collected to describe the population reached and to include clinical characteristics (e.g., cancer type and treatment status and type), demographics (e.g., age, gender, ethnicity, education status, and work status), and location (rural/remote vs. urban communities). Enrolment represents the number of study participants after screening and completion of consent by the hub CEP. Reasons for study refusal were collected if a potentially interested participant did not enrol in the study. Referral types included “direct HCP referral”, “indirect HCP referral”, or “self-referral”. Direct HCP referral included an HCP directly contacting a hub CEP about a potential participant. Indirect HCP referrals involved a participant contacting a hub CEP after receiving information about EXCEL from an HCP. Self-referrals included participants contacting a hub CEP after hearing about EXCEL from other sources (e.g., current/former participant, support groups, or social media). Finally, referral resources were determined upon screening by the CEP. Participants were asked “How did you first find out about the EXCEL program?” to better understand the referral resources they received. Referral resources are categorized as ”Print Materials”, ”EXCEL Team Outreach”, “Word of Mouth”, or “HCP”. 

#### 2.4.2. Adoption 

##### Healthcare Provider Network

Hub CEPs led outreach to rural and regional cancer centres, hospitals, primary care networks, universities/research teams, and community healthcare clinics (e.g., cancer physiatry clinics) to build the HCP network. HCPs were able to join the EXCEL network at any time, either via our outreach or on their own accord. HCPs included oncologists, physiatrists, primary care physicians, nurses, social workers, physiotherapists, registered dieticians, patient navigators, and care coordinators. Outreach efforts include educational presentations, emails, and phone calls to share resources and information. Specifically, HCP in-person and online education sessions were offered to disseminate general exercise-oncology information (i.e., exercise guidelines and currently available exercise oncology resources) and EXCEL-specific referral information. Regular email outreach included emailing cancer centre contacts with recruitment materials for EXCEL, including posters and brochures (see Appendix A). Email outreach to the HCP network occurred six weeks prior to a new EXCEL program, with additional emails sent every two weeks until the program start date to remind them about the referral process (see sample referral form in Appendix A). Finally, phone calls were also made to HCPs at least twice a year, providing information about EXCEL and addressing any referral needs. Markers of adoption in building the HCP network included the number and type of HCPs (i.e., HCP professional/clinical role and organization) and details from the HCP organizations that received educational/referral materials (i.e., brochures and information sessions provided by the EXCEL team). 

##### Qualified Exercise Professional Network 

The second EXCEL network essential to establishing our sustainable community-based exercise-oncology program was the QEP network. This network included QEPs from primarily rural/remote communities who had the capacity to deliver the online exercise intervention. QEPs from urban centres may have also been part of the network, in particular to deliver to those who had lost access to their usual in-person classes during COVID-19 restrictions. QEPs were required to have a background in kinesiology, preferably with exercise certifications from accredited organizations. This included the Canadian Society of Exercise Physiology (CSEP) Clinical Exercise Physiologist (CEP) or Certified Personal Trainer (CPT) certifications, the American College of Sports Medicine (ACSM) CEP, CPT, Exercise Physiologist, Cancer Exercise Trainer certifications, Registered Kinesiologists (R.Kin), and group fitness instructors certified through the National Fitness Leadership Alliance (NFLA), National Council on Strength & Conditioning, Canfitpro, YMCA Canada, or ACSM. 

Once QEPs expressed an interest in EXCEL, a comprehensive training pathway was initiated. Specifically, QEPs in the EXCEL network completed exercise-oncology and behaviour-change education and training, providing a foundation to safely deliver online exercise classes to individuals living with and beyond cancer. Educational materials included online (approximately 15 h; asynchronous learning) cancer- and exercise-specific trainings from Thrive Health Services (www.thrivehealthservices.com, accessed on 30 November 2022). Module topics provided detailed information about screening/assessment, cancer and exercise prescription, psychosocial considerations, and health behaviour changes. An EXCEL-specific training day (online; synchronous learning) was also provided on additional behaviour-change topics, study logistics (including safety protocols), and exercise-program delivery expectations to ensure consistency across EXCEL implementation sites. Lastly, QEPs were required to moderate (i.e., act as a secondary instructor) online exercise classes so that they had an opportunity to become familiar with online exercise-program delivery prior to being the primary instructor of a class. Markers of adoption for the QEP network included the number of QEPs trained to deliver EXCEL, the number of QEPs actively delivering EXCEL exercise classes, and both the number and type of fitness partnership that implemented EXCEL in the first year (e.g., individual QEPs or health and wellness organizations including fitness centres, fitness partners through healthcare settings, or other organizations).

#### 2.4.3. Implementation

Implementation of the EXCEL program included study feasibility and safety of the exercise intervention delivery. Specifically, feasibility included the retention rate, adherence rate, reasons for missing exercise classes, assessment completion rate, and total number of exercise classes provided. On the other hand, safety included the reporting of adverse events. The retention rate represents the percentage of participants that completed the 12-week intervention and associated study assessments. Adherence is defined as the total number of exercise classes attended from the total number of possible classes (20–24 classes, depending on participant start date). Reasons for missing exercise sessions were also tracked to inform the study team of potential barriers. Reasons included technology-related issues (e.g., WiFi or ZOOM™ link issues), class conflicts (e.g., appointments, working, traveling), cancer-related issues (e.g., symptoms, treatment appointments), or non-cancer-related health issues unrelated to EXCEL participation (e.g., illness or injury). Provided that the education webinars were an optional part of EXCEL and could be attended live on ZOOM™ or watched via recording, adherence to the webinars was not tracked. Completion rates for the fitness assessment and patient-reported outcomes at baseline and 12-week timepoints were also tracked. Patient-reported outcomes included measures of fatigue, quality of life, health status, symptom burden, barriers and facilitators, and physical activity. Reasons for not completing fitness assessments at baseline and 12-week timepoints were tracked and categorized into the following: unable to contact or schedule after repeated attempts; cancer-related or non-cancer-related physical limitation; declined participation in fitness assessments; and no reason provided. The total number of exercise classes provided per hub was tracked to better understand both the demand for online exercises classes across regions and to track the number of QEPs delivering classes at sites as a marker of the personnel required to deliver the intervention. Safety included tracking the number and grade of reported adverse events via the Common Terminology Criteria for Adverse Events (CTCAE) Version 5.0 form [21]. 

### 2.5. Analysis

Participant demographics, clinical characteristics, and all reach, adoption, and implementation outcomes were reported using descriptive statistics including the mean, range, standard deviations, or frequencies and percentages, as appropriate. All data were collected and stored in REDCap (Research Electronic Data Capture) [22,23] and exported to RStudio Version 1.3 (Boston, MA, USA), where descriptive analyses were performed.

## 3. Results

All summary results for reach, adoption, and implementation are presented in Figure 2 (EXCEL First Year Enrolment), Figure 3 (EXCEL regional HCP networks), Table 1 (participant demographic and clinical characteristics), Table 2 (reach, adoption (QEP), and implementation summary), Table 3 (referral resources), and Table 4 (presentations to HCP network). All reach, adoption, and implementation outcomes in Table 2 were reported as totals for year one and grouped by each hub site (i.e., Alberta, Nova Scotia, and Ontario). Characteristics, reach, and implementation outcomes grouped by participant location (i.e., rural/remote or urban) can be found in Appendix A.

### 3.1. Reach

#### 3.1.1. Participant Demographics and Clinical Characteristics 

Demographics and clinical characteristics (Table 1) show that the EXCEL participants in year one were primarily from rural/remote areas (84.3%), of British or European descent (64.8%), female (81.4%), retired (36.0%), had at least a college education or equivalent (88.6%), and were either married or reported having a common-law partner (75.8%). Clinically, most participants had a breast cancer diagnosis (53.4%) and were actively receiving treatment (54.2%), the most common of which was hormone therapy (40.6%). Slightly more than a quarter of the participants in year one reported having an advanced cancer diagnosis (27.1%). 

#### 3.1.2. EXCEL Study Enrolment, Study Refusal, Referral Type, and Referral Resources

Over the first year of the EXCEL study, there were a total of 338 individuals living with and beyond cancer who expressed interest. A total of 290 individuals provided informed consent and enrolled in the study (see Figure 2). Reasons for non-enrolment (*n* = 48) included the following: unable to be contacted after repeated attempts (*n* = 36), no longer being interested in the study (*n* = 7), and being deemed ineligible after screening (*n* = 5). Self-referrals were the predominant referral type to the EXCEL study in the first year (*n* = 179, 75.8%), compared to indirect HCP referrals (*n* = 37, 15.7%) and direct HCP referrals (*n* = 20, 8.5%). Participants reported hearing about the EXCEL study through health and wellness programs or organizations (*n* = 77), support groups (*n* = 34), and from previous/current participants in the study (*n* = 21). HCP referrals primarily resulted from interactions and conversations with nurses (*n* = 16) and oncologists (*n* = 12) about the EXCEL study. A complete breakdown for all referral types and resources, including print materials (i.e., brochures and posters) and EXCEL team outreach (i.e., online presentations, social media posts, and study staff outreach) are recorded in Table 2 and Table 3. 

### 3.2. Adoption

#### 3.2.1. Healthcare Provider Network

As a result of outreach presentations, emails, and phone calls to HCPs, the EXCEL study established 163 clinical contacts across the three hubs, which included cancer centres, primary care networks, hospitals, universities, and community clinics. The EXCEL team provided 14 outreach presentations tailored for HCPs to 11 different organizations in Canada in year one. Organizations included cancer centres (*n* = 4), cancer care organizations (*n* = 5), and non-profit organizations (*n* = 2). Details regarding the various organization presentations can be seen in Table 4. In total, the Alberta hub established 55 clinical contacts, Nova Scotia established 21, and Ontario established 87. The Alberta hub also established clinical contacts from other provinces and territories within Canada during year one, including British Columbia, Saskatchewan, and the Northwest Territories. This resulted in *n* = 13 participants from British Columbia and *n* = 23 participants from Saskatchewan. Regional clinical contacts between each of the three hubs can be seen in Figure 3 (EXCEL regional HCP networks). Please refer to Appendix A for specific HCP organizations (e.g., cancer centres, primary care networks, and hospitals) associated with each hub. 

#### 3.2.2. Qualified Exercise Professional Network 

A total of 32 QEP partners were established within EXCEL in the first year. Health and wellness organizations with multiple QEPs accounted for 19 of the partnerships, and 13 partnerships were established with individual QEPs. In total, 45 individual QEPs were trained to effectively deliver evidence-based exercise-oncology programs. Of the 45 trained QEPs, 22 actively delivered EXCEL online exercise classes in year one. Table 2 provides further details regarding the QEP network and is grouped by hub. 

### 3.3. Implementation

Of the 290 enrolled participants, 236 completed the 12-week exercise oncology program (retention rate = 81.4%). Reasons for study drop-out included “no longer being interested” (*n* = 18), “personal and/or health reasons impacting participation” (*n* = 11), or “other” (*n* = 25), which included reasons such as “questionnaire burden”, “time commitment”, or “work/schedule conflicts.” Combined, adherence to the online exercise intervention in year one was 78.2%. Adherence varied across the three hubs (Alberta = 76.7%; Nova Scotia = 82.3%; and Ontario = 77.0%) as well as across enrolment periods (Fall 2020 = 84.3%, Winter 2021 = 84.4%, Spring 2021 = 72.4%, and Fall 2021 = 76.9%). The majority of participants missed exercise classes due to class conflicts (36.4%), followed by non-cancer-related health issues (18.0%), cancer-related issues (7.3%), technology-related issues (2.1%), or no reason was provided (36.3%). 

Baseline and 12-week online questionnaires of patient-reported-outcome completion rates were 96.6% (228/236) and 87.7% (207/236), respectively. Reasons for not completing online questionnaires were not tracked. Baseline and 12-week fitness-assessment completion rates were 98.3% (232/236) and 89.8% (212/236), respectively. Reasons for not completing baseline and 12-week fitness assessments (*n* = 28) included being unable to contact or schedule after repeated attempts (*n* = 12), cancer-related or non-cancer-related physical limitations (*n* = 7), declined participation (*n* = 3), and no reason provided (*n* = 6). A total of 37 12-week EXCEL exercise classes were delivered in year one (Alberta = 16; Nova Scotia = 14; and Ontario = 7). With hubs established throughout Canada contributing to the growing number of trained QEPs in the first year, the number of exercise classes provided also increased with each additional cohort (Fall 2020, three 12-week exercise classes offered via the Alberta hub; Winter 2021, ten 12-week exercise classes offered via the Alberta and Nova Scotia hubs; Spring 2021, eleven 12-week exercise classes offered via all three hubs; and Fall 2021, thirteen exercise classes offered via all three hubs). Only one adverse event was reported in the first year, due to potential over-exertion with associated fatigue and headache after finishing an exercise circuit. This event was deemed a “minor incident with no lost time” (‘mild’—Grade 1) [21].

## 4. Discussion

In year one, the EXCEL study established a Canada-wide exercise-oncology network including three hubs in Alberta, Nova Scotia, and Ontario, facilitating outreach and implementation with an HCP network of 163 clinical contacts and a QEP network of 45 trained fitness professionals. The HCP network primarily supported referral and building awareness of the role of exercise oncology across clinical settings, while the QEP network enabled the safe and effective delivery of the EXCEL intervention. Together, these networks provided an opportunity for 338 individuals living with and beyond cancer, from both rural and remote locations (84.3%) and urban locations (15.7%), to participate in the online EXCEL study. Adherence and study assessment completion rates and the number of reported adverse events also supported that EXCEL was both feasible and safe in year one. These results provide a preliminary insight into how the EXCEL study has begun to effectively address access to exercise-oncology programs for individuals living primarily in rural and remote locations, or to those in urban settings when access to exercise-oncology resources is disrupted. These findings also highlight the continued needs that will be addressed throughout the five-year study. 

### 4.1. Reach and Adoption

Through development of HCP and QEP networks, EXCEL established its reach and adoption in year one. Enrolment increased from 27 participants in the first wave of participation (Fall 2020) from the Alberta hub to enrolling 103 participants in the Fall of 2021 with the Alberta, Nova Scotia, and Ontario hubs actively recruiting. While EXCEL has been successful to date in reaching individuals in rural/remote settings, the present sample reflected what is commonly observed across exercise-oncology studies, with most participants reporting a breast cancer diagnosis (53%) and no other tumour group making up more than 8.5% of the sample (lung cancer comprised 8.5%; hematologic cancer comprised 8.1%). This finding underscores a clear need to expand directed outreach to other tumour populations and consider the tailoring of exercise-oncology resources to ‘meet their needs’. For example, EXCEL has started to offer tailored, lung-cancer-specific online exercise classes in response to a request from participants and with outreach support from Lung Cancer Canada [24]. The tumour-specific EXCEL classes represent a tailored and accessible exercise resource that will specifically target unmet needs in individuals living with lung cancer. Since targeted outreach began, there have been 22 participants in two successful, lung-specific exercise classes in 2022. This targeted outreach and delivery of classes will be expanded to reach other underserved tumour types, including for individuals with an advanced cancer diagnosis that require specific exercise tailoring to meet functional and movement needs [25,26]. 

In addition, EXCEL participants to date have largely demographically been consistent with prior research. For example, very few participants in the EXCEL study self-identified as a member of an ethnic minority population (10.5%), reported an education of high school or less (10.7%), were male (17.9%), or would be classified as “young adults” (6.8%), defined as being between the ages of 18 and 39 years old. Given the need to ensure all individuals living with and beyond cancer must be included within exercise oncology interventions, ongoing work examines how to tailor resources (education and programming) to be more inclusive across all sub-populations to improve access [27,28,29]. In addition, ongoing EXCEL initiatives include the translation of documents into French to ensure access to French-speaking individuals with cancer, as well as seeking additional support for partnering with Indigenous communities to co-develop tailored exercise-oncology resources. 

Most of the year-one sample were currently receiving cancer treatment (54.7%) while participating in the EXCEL online exercise intervention. This is an important finding, given the consistent reports of reduced physical activity that often occurs across the cancer continuum after a cancer diagnosis, and especially during treatment, when both physical and psychosocial burdens may be high [30,31,32]. The success of EXCEL implementation with individuals living with cancer while receiving treatment may be indicative of the “meet patients where they are at” approach within EXCEL, the support provided by the highly trained QEPs to modify and support each individual within the online exercise setting, and the increased awareness of the role of exercise across the clinical setting (i.e., the success of HCP outreach and referral). Further qualitative work with participants will elucidate both facilitators and barriers for those on-treatment to ensure that EXCEL continues to address a critical need of providing exercise resources during treatment within the cancer care system. 

EXCEL’s reach also involved developing simple referral pathways for HCPs that effectively connected rural and remote individuals living with cancer with hub CEPs for screening and enrolment into EXCEL. This referral pathway process varied across regions, with indirect HCP referrals predominantly seen from the Alberta hub while direct HCP referrals were predominantly seen from the Nova Scotia hub. This finding may reflect differences in outreach and established contacts in various regions across Canada. For example, the Alberta hub established clinical contacts with 55 HCP organizations, resulting in 33 indirect HCP referrals. Nova Scotia established clinical contacts with 21 HCP organizations, resulting in 18 direct and 2 indirect HCP referrals (50% of their referrals to EXCEL). Comparatively, the differences in referral types and the number of clinical contacts highlighted different outreach strategies across regions. In Alberta, this may indicate that HCPs are providing their patients with recruitment and referral resources given to them from hub CEPs (i.e., resource uptake), which in turn results in indirect HCP referrals. In Nova Scotia, there is a clear communication pathway between HCPs and hub CEPs, contributing to the increased percentage of direct HCP referrals. Regardless, these findings are encouraging and suggest the possibility of building HCP support for exercise-oncology referral and participation. In turn, this contributes to increased accessibility to exercise-oncology resources to support wellness in cancer survivorship across regions and healthcare networks. 

Despite the promising development of EXCEL’s HCP referral pathways, most participants (57.7%) heard about EXCEL via “word of mouth” resources and self-referred into the study. This is not surprising, given commonly reported HCP barriers to exercise-oncology referral, including a lack of time, knowledge, and resources about exercise and cancer [33], as well as limitations to in-person interactions associated with the COVID-19 pandemic (i.e., less facetime between HCPs and patients in clinical settings). Although enrolment through self-referral highlighted the effective community-based outreach from the EXCEL hubs, the HCP referral pathway is critical for sustainability within the cancer care system [14,25]. Ongoing work in both neuro-oncology [34] and head and neck cancer [35] populations are utilizing rehabilitation triage clinics in collaboration with physiatrists within cancer care settings to embed both exercise screening and referral into clinical workflows. This work, in addition to the continued development of the HCP referral pathways that EXCEL established across regions, will directly address recent calls to action for integrating exercise-oncology resources into cancer care [25,36,37]. 

To continue to build upon established HCP referral pathways, EXCEL outreach plans for the HCP network include continuing to address common HCP barriers to exercise referral (i.e., time and knowledge) via simplifying referral processes, including building an online referral process. In addition, the EXCEL team has developed an “Exercise Oncology Postcard” that is designed to be an easy-to-read resource with exercise-oncology resources and contact information. This is distributed to HCPs in various clinics for sharing with their patients for referral to EXCEL. Ongoing educational work with HCPs to support exercise as part of standard cancer care, in addition to advocating with the administration in clinical settings, will support establishing exercise referral as part of electronic medical-record systems within cancer care, ensuring exercise-oncology resources for patients are easily accessible and thus remain an evidence-based sustainable component of the cancer care system [14,33]. 

The QEP network development is also essential to ensure access to a qualified and trained fitness professional who can effectively deliver exercise-oncology resources for individuals living with and beyond cancer. The EXCEL QEP network engagement was successful in the first year, supporting EXCEL awareness across rural and remote communities connected to the three first-year hubs, as well as supporting the implementation of the EXCEL intervention. The QEP network included individual fitness professional partners as well as partnerships with health and wellness organizations (e.g., Wellspring, WE-Can, YMCA). With the support of the EXCEL research team, exercise-oncology training, education, and resources were provided to the QEP network members (*n* = 45 QEPs), resulting in 22 QEPs being involved in the delivery of 37 EXCEL online exercise classes. 

To our knowledge, EXCEL is the first study to report on QEP network development through rural/remote fitness partnership outreach and training. With recent calls to develop an “exercise oncology workforce” to meet the physical activity and exercise needs of millions of individuals living with and beyond cancer each year [36], our results provide a timely insight as to how institutions can feasibly develop a QEP network to safely and effectively deliver online exercise-oncology programs to individuals in rural and remote communities. Importantly, in building the EXCEL QEP network, not all trained QEPs actively delivered EXCEL exercise classes (i.e., 22 delivering vs. 45 total trained), as delivery was based on the participant demand for classes; however, all QEPs are still considered important partners within EXCEL. The QEP network is provided with ongoing, free exercise-oncology and behaviour-change education (e.g., webinars, educational events, and EXCEL newsletters), and can utilize their training within their communities, ultimately providing exercise oncology resources within any fitness setting they work in. 

To further develop EXCEL’s QEP network, direct outreach to major fitness professional organizations will occur. For example, EXCEL recently promoted the QEP training pathway to the Canadian Society of Exercise Physiology (CSEP) members to recruit QEPs in rural and remote communities, and relationships with the YMCAs and various health and wellness organizations in Nova Scotia and Ontario are supporting growth across regions. Outreach of this kind will be critical for the future implementation of EXCEL, particularly when in-person exercise classes are offered. The sustainability of EXCEL’s QEP network will be continually evaluated as well. QEPs that underwent the EXCEL training were approached to complete follow-up surveys and provide information regarding if and how they utilize their training in practice (i.e., delivering EXCEL classes or delivering their own exercise and cancer classes), providing valuable information about the utility and needs for continuing education and engagement of our exercise-oncology QEP network. 

### 4.2. Implementation

Implementation of the EXCEL study in the first year was also promising. Compared to previous work, EXCEL’s exercise intervention adherence rate (78%; a range of 76–82% across regions) was lower than the adherence rates of recently published online exercise-oncology interventions of 86–91% [7,8]. However, the unique timing related to the COVID-19 pandemic was a critical factor. Both prior studies [7,8] indicated that the online exercise interventions were completed during early lockdowns in the pandemic, potentially contributing to the increased adherence rates as participants had the time to exercise at home. This trend was observed in EXCEL, with the highest adherence rates being reported in earlier enrolment periods (Fall 2020 (84.3%) and Winter 2021 (84.4%)), whereas lower adherence rates were reported during periods in which lockdowns began to vary across regions (Spring (72.4%) and Fall 2021 (76.9%)). The top reason for missing classes was due to general class conflicts (36.4%), which included reasons such as work, travel, or family obligations. It was encouraging to see that very few participants across regions reported technology as being a reason for missing an exercise class (2.1%), particularly as many individuals were from rural and remote regions where remote access (i.e., WiFi) may be limited [38]. To better understand the feasibility of the EXCEL exercise intervention, reasons for not attending exercise classes will continue to be tracked. This will increase our understanding of the barriers that may limit those living with cancer from rural and remote communities from attending online exercise classes and how these can be addressed across regions for EXCEL implementation in years two to five. Supporting higher adherence rates to the EXCEL intervention, in order to optimize the potential exercise benefits for participants, will be critical to ensuring that online delivery is both sustainable and effective for individuals living with and beyond cancer in rural and remote regions.

The completion rates of the online fitness assessments and patient-reported outcomes were greater than 85% at both baseline and 12-week timepoints. Improving these completion rates to be greater than 90% will be a key focus for the EXCEL team, as the data collected from this trial will have important implications for the effectiveness of exercise-oncology interventions in real-world settings. In particular, it will provide novel data for online, supervised group-exercise-intervention delivery. Using the REDCap database [22,23], data quality checks were completed by an EXCEL study team member, and timed reminders to complete assessments were sent electronically (i.e., via email) to participants. In addition, QEPs reminded participants during EXCEL sessions about scheduling upcoming post-intervention (12-week) assessments. 

### 4.3. Future Directions: EXCEL Implementation Strategies

To support EXCEL implementation as “best-evidence to best-practice”, efforts towards sustainable network development are ongoing. For example, EXCEL will continue to develop and maintain clinical referrals via the development of the HCP network, and QEP partnerships will continue to grow to ensure both the online and in-person intervention delivery reaches as many potential participants as possible. Additional hubs in other regions across Canada will be added (i.e., British Columbia), which will contribute significantly to HCP and QEP network development. Support will be provided to build sustainable contacts through continual HCP outreach (i.e., ~6-weeks prior to each new program start date) and by offering free educational sessions, such as grand rounds presentations or brief nursing huddles to potential and current clinical contacts, providing referral materials such as study brochures/posters and exercise-oncology postcards, distributing a monthly newsletter that provides updates on the overall study progress, and providing direct contact via online referral systems to hub CEPs. Additionally, with the lifting of COVID-19 restrictions allowing for face-to-face interactions in tumour clinics, increased face time between EXCEL study team members and HCPs within clinical settings will also facilitate both indirect and direct HCP referrals. With these strategies in place, EXCEL will continue to build the HCP network that supports referral across tumour groups. Building participation that expands to include additional tumour types, ages, more males, and ethnic minority populations will ensure that the EXCEL study will inform future research regarding the real-world implementation of an exercise-oncology intervention that works for more diverse cancer populations in rural and remote communities. Finally, the lifting of COVID restrictions and thus the ability for urban centres to deliver exercise-oncology programs also means a focus within EXCEL upon reaching rural/remote participants, with urban participant inclusion only if there is no ongoing access to in-person or online programs.

Ongoing work also includes quality improvement surveys distributed in six-month cycles (i.e., QI cycles) to both HCP and QEP networks as well as participants. QI surveys are sent via email to HCPs and QEPs to gather information about the level of support they receive from the EXCEL research team, the feasibility of study implementation related to referrals and delivery of the program, and suggestions that may allow them to feel better supported to adopt the EXCEL program and maintain their roles. Similarly, participants provide feedback on study components including fitness assessments, educational materials, the exercise program, and their overall experience. With the first year of EXCEL now complete, QI survey feedback will begin to be reviewed to identify pertinent “areas of improvement” and gaps in support that need to be addressed. These surveys, in addition to semi-structured interviews, will provide real-time feedback to the EXCEL research team, allowing for adjustments that will sustain and support the developed HCP and QEP networks, and will, in turn, support sustainability for access to exercise-oncology resources for rural and remote individuals living with and beyond cancer.

## 5. Conclusions

The delivery and implementation of the EXCEL study in year one was considered feasible and safe, with implementation largely facilitated through the development of HCP and QEP networks. The outreach and established HCP network of the EXCEL study supported the increase in program referrals predominantly from rural and remote participants throughout year one. All EXCEL QEPs received exercise-oncology and behaviour-change training, providing them with the background necessary to deliver EXCEL online exercise classes or to apply the learned skills within their own communities if not actively delivering EXCEL-specific online exercise classes. In turn, the developed QEP network contributed to the safe and feasible delivery of the EXCEL online-exercise intervention. These findings provide important information related not only to developing supportive networks for exercise-oncology delivery for those in rural and remote areas, but also to strategies for sustainable exercise-oncology implementation. EXCEL will continue to employ “best-practice” implementation strategies throughout the five-year study to ensure that HCPs and QEPs are supported to adopt and maintain community-based exercise-oncology programs so that those living with cancer in rural and remote areas can access, feasibly participate in, and experience the physical and psychosocial benefits of exercise.

## Figures and Tables

**Figure 1 ijerph-20-01930-f001:**
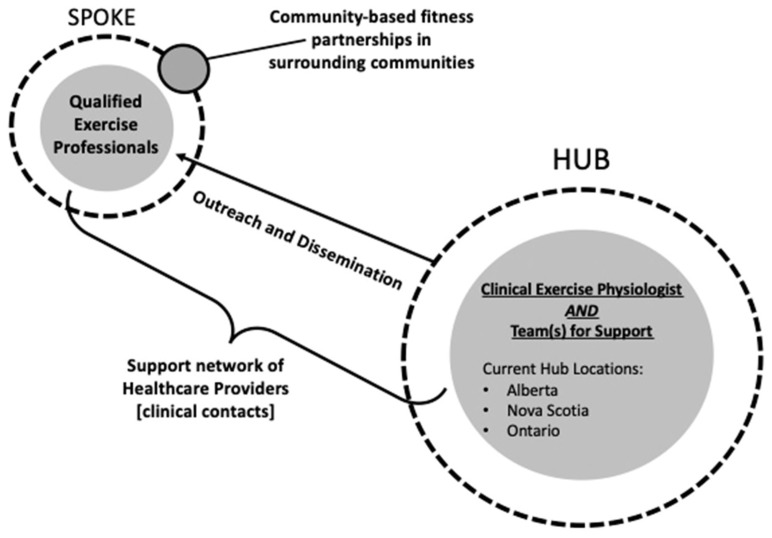
EXCEL “hub and spoke” framework.

**Figure 2 ijerph-20-01930-f002:**
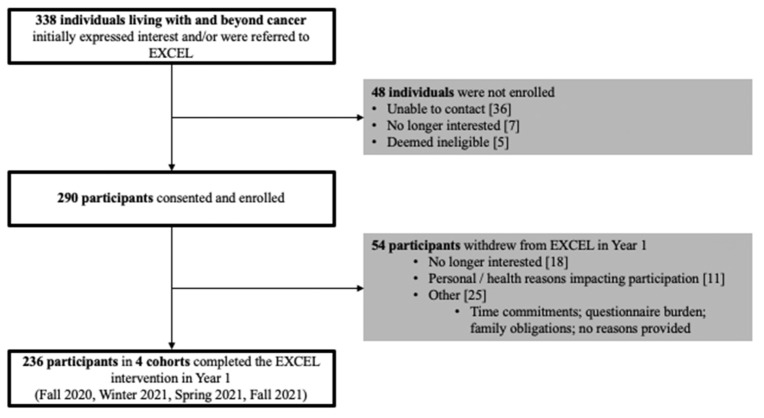
EXCEL first-year enrolment.

**Figure 3 ijerph-20-01930-f003:**
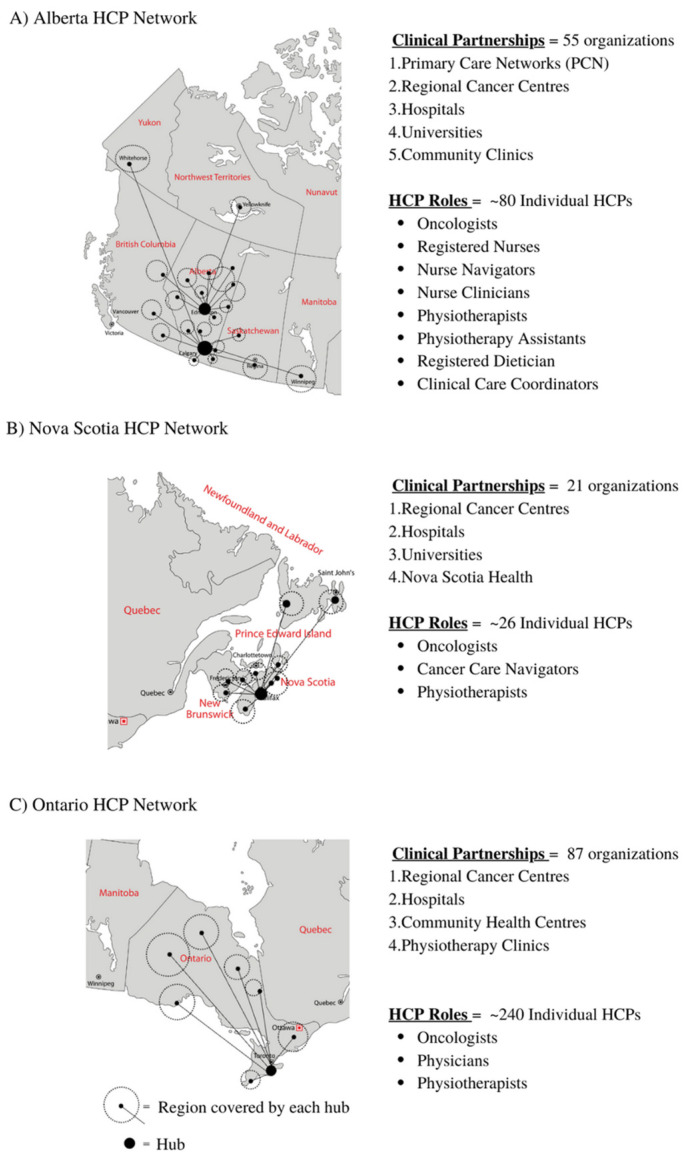
(**A**–**C**): EXCEL regional HCP networks.

**Table 1 ijerph-20-01930-t001:** EXCEL participant demographics and clinical characteristics—number (%).

Variable	Participants (*n* = 236)
Age (years) ^†^	57.3 (12.5)
Location	
Rural/Remote	199 (84.3)
Urban	37 (15.7)
Sex	
Female	192 (82.1)
Male	42 (17.9)
Ancestral Ethnicity	
Aboriginal	4 (1.7)
Asian	14 (5.9)
British	81 (34.3)
Caribbean	1 (0.4)
European	72 (30.5)
Latin/Central and South America	6 (2.5)
Multi-ethnic	38 (16.1)
No information provided	20 (8.5)
Education	
High school or less	25 (10.7)
More than high school	209 (89.3)
Employment Status	
Full Time	41 (17.5)
Part Time	31 (13.2)
Retired	85 (36.3)
Homemaker	8 (3.4)
Disability	56 (23.9)
Temporarily Unemployed	13 (5.6)
Marital Status	
Not Married	55 (23.5)
Married	152 (65.0)
Common Law	27 (11.5)
Cancer Type	
Breast	126 (53.4)
Lung	20 (8.5)
Hematologic	19 (8.1)
Gynecological	16 (6.8)
Prostate	16 (6.8)
Skin	6 (2.5)
Brain	6 (2.5)
Colon	6 (2.5)
Head and Neck	5 (2.1)
Other ^‡^	16 (6.8)
Advanced Cancer	
Yes	64 (27.4)
No	170 (72.6)
Treatment Status	
On	128 (54.7)
Off	106 (45.3)
Treatment Type for participants ‘On’ treatment	
Chemotherapy	33 (26.2)
Radiation	5 (4.0)
Hormone Therapy	52 (41.3)
Combination *	16 (12.7)
Other **	20 (15.9)

^†^ = Age is expressed as the mean (standard deviation). ^‡^ Other = bladder, appendiceal, kidney, pancreatic, small bowel, colorectal, gastric, rectal, and esophageal. * Combination = any combination of chemotherapy, radiation, hormone therapy, or “other.” ** Other = immunotherapy, targeted therapy, surgery, or antimetabolites.

**Table 2 ijerph-20-01930-t002:** Reach, adoption (QEP), and implementation results.

Variable (*n*, % of Year One Total)	Alberta	Nova Scotia	Ontario	Year One Total
Reach (*n,* % of Year one total)				
Enrolled Participants	113	60	63	236
Direct HCP Referral	0	18	2	20 (8.5%)
Indirect HCP Referral	33	2	2	37 (15.7%)
Self-Referral	80	40	59	179 (75.8%)
Adoption—QEP Network (*n*)				
Fitness Professional Partnerships	13	7	12	32
Health and Wellness Organizations	4	6	9	19
Individual QEPs	9	1	3	13
Trained QEPs	17	12	16	45
QEPs Delivering EXCEL Classes	9	6	7	22
Implementation (*n* or %)				
Exercise Intervention Adherence	76.7%	82.3%	77.0%	78.2%
Fitness-Assessment Completion				
Baseline	100.0%	93.3%	100.0%	98.3%
12-week	92.0%	90.0%	85.7%	89.8%
Patient-Reported Outcome Completion				
Baseline	99.1%	96.7%	92.1%	96.6%
12-week	92.0%	90.0%	77.8%	87.7%
Exercise Classes Implemented	16	14	7	37

**Table 3 ijerph-20-01930-t003:** Referral resources *.

Referral Resource Descriptions	Number of Times Resources Were Indicated
Word of Mouth (57.7%)	
Friend or Family Member	3
Previous or Current Participant	21
Support Group	34
Health and Wellness Organizations and Programs	77
EXCEL Team Outreach (20.1%)	
Online Presentations	1
Social Media or Websites	19
Study Staff	27
Healthcare Provider (18.8%)	
Social Worker	1
Lymphedema Specialist	1
HCP provided brochure in clinic	3
Patient Navigator	3
Care Coordinator	3
Primary Care Network	5
Oncologist	12
Nurse	16
Print Materials (3.4%)	
Posters	3
Brochures	5

* = participants could indicate multiple referral resources if they chose to answer the question “How did you first find out about the EXCEL program?”

**Table 4 ijerph-20-01930-t004:** Presentations to HCP network.

HCP Organizations (*n* = 11)	Number of Presentations (*n* = 14)	Hub That Provided Presentation
Cancer Centres		
Tom Baker Cancer Centre	3	Alberta
Allan Blair Cancer Centre	1	Alberta
Margery E. Yuill Cancer Centre	1	Alberta
Jack Ady Cancer Centre	1	Alberta
Cancer Care Organizations		
Nova Scotia Health (Medical Oncology)	1	Nova Scotia
Nova Scotia Cancer Care Clinic	1	Nova Scotia
Canadian Association of General Practitioners in Oncology	1	Nova Scotia
Canadian Association of Psychosocial Oncology	1	Alberta
Cancer Care Ontario	1	Ontario
Non-Profit Organizations		
Exercise is Medicine	2	Alberta
Inpower	1	Alberta

## Data Availability

Data can be made available upon reasonable request.

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
