# Peer review of "First-Year Implementation of the EXercise for Cancer to Enhance Living Well (EXCEL) Study: Building Networks to Support Rural and Remote Community Access to Exercise Oncology Resources"

_ijerph, 2023, doi:10.3390/ijerph20031930_

Round 1
Reviewer 1 Report
General
Wagoner and colleagues present a well written manuscript describing the 1-yr intermediary results of the EXCEL Study, which aims to support oncology patients via exercise programs especially in rural and remote communities across Canada. The authors utilized the RE-AIM framework (i.e., Reach, Effectiveness, Adoption, Implementation, and Maintenance which together are thought to determine public health impact).
Wagoner and colleagues report very high adherence and completion rates, feasibility and successfully implemented networks between patients, health care professionals and qualified exercise professionals. Readers do not get any substantial information regarding the effectiveness of the intervention (e.g., changes in physical fitness, socio-economic burden or benefits).
Critique
From my point of view, three things should be criticized.
1st, the rather lengthy discussion.
2nd, the impression that some of the intentions planned for the future, such as "ongoing integrated knowledge translation efforts" and "feedback from stakeholder groups", do not result from the reported data, but were probably planned a priori, completely independently of the results. I do not want to criticise the importance and correctness of these plans, but according to my scientific understanding, they do not belong in the paper. Even in such a methodological paper, the conclusion should be data-based.
3rd, I am missing data on the socio-economic aspects, at least the costs of the intervention per patient should be summarized and ideally compared to the costs of a non-digital exercise intervention.
Minor
Before recommending the acceptance of the paper, I have a few minor comments that I respectfully request the authors to include in their reflections:
Abstract
L20: Please specify the barriers you have in mind here, similar to L 46 (geographic isolation etc.). We know there are several barriers hindering peoble to participate in rehab and exercise programs.
L24: I think you should either briefly explain the RE-AIM concept in the abstract, or avoid the term.
L35: See above, the “conclusion” is not data supported.
L38: Please avoid repeating keywords from the title
Introduction
L42: You might want to add the information that cardiovascular fitness is linked to mortality / survival probability with or without treatment (e.g., https://www.mdpi.com/2072-6694/13/8/1771)
L57: A strong argument for remote rehabilitation etc. are potential socio-economic benefits. Please consider naming this aspect here.
Materials
L104: I apologise for not being familiar with the Canadian system, however, as a reader, I wonder if the definition of "underserved rural/remote communities (population <100,000 people)" is generally acceptable. 100,000 people would make a small but complete city in Central Europe. Please consider to base your definition on population density. On Wikipedia I found “a population density of less than 150 people per square kilometer” defining rural areas in Canada. https://en.wikipedia.org/wiki/Rural_area#Canada
Results
Fine. Thank you.
Discussion
I feel the discussion is overlong and would politely ask you to consider the possibility of compressing it.
Reviewer 2 Report
Please see attached file
